# OpenReview forum: "HighClass: Efficient Metagenomic Classification via Quality-Aware Token Mapping and Sparsified Indexing"
_ICLR.cc/2026/Conference — ICLR 2026 Conference Withdrawn Submission_

### Official Review · Reviewer_tfiW · 2025-10-26

**Soundness:** 2
**Presentation:** 2
**Contribution:** 2
**Rating:** 2
**Confidence:** 4

**Summary:**

The paper addresses the metagenomics classification problem, where the goal is to assign short DNA fragments (known as reads) to taxonomic categories such as species, family, or class. Instead of relying on computationally expensive sequence alignment techniques, the authors propose a hash-based token mapping framework that aims to achieve an improved trade-off between computational efficiency and classification performance. The approach is evaluated on the CAMI II dataset and demonstrates higher throughput than baselines.

**Strengths:**

- The proposed approach shows improved throughput compared to alignment-based baselines.
- The inclusion of a theoretical analysis is valuable for providing deeper insights into model behavior and guarantees.

**Weaknesses:**

- The presentation and organization of the paper are not good. Especially for readers unfamiliar with the metagenomics classification problem,  it is difficult to follow the paper.
- The paper provides a limited novelty. The proposed approach mainly relies on MetaTrinity, with some modifications (e.g., hash-based mapping).
- Although the authors emphasize that the theoretical analysis is a major contribution of the work, it is discussed in the appendix and presented incompletely.
- The experiments are limited to the CAMI II dataset. The baseline method performs better in accuracy than the proposed one (though slower).
- The paper omits and insufficiently discusses key related methods.
- The appendix is incomplete.

**Questions:**

- Several key baselines, including Metalign and MetaTrinity, are missing from Tables 2 and 4. Could the authors explain why they were omitted?
- The paper does not specify how the proposed approach handles reads that originate from unknown taxa, i.e., organisms absent from the reference database. Could the authors clarify this behavior for assessing the robustness and applicability of the method in realistic scenarios?
- The description of hyperparameter tuning is vague. Section M.3 mentions a sensitivity analysis, but no corresponding results or experiments are provided. Could the authors elaborate on how hyperparameters were selected and provide supporting experimental evidence?
- The discussion of related work should be expanded to include key methods such as MetaCache-GPU, MetaBinG2. In addition, the baseline methods, MEGAN, and Metalign, should be cited.
- In the appendix, several sections (F.1–F.3, M.5, M.7) are empty, and references between Propositions and proofs are mismatched (e.g., Proposition 4 vs. Proposition 20). The proofs of Theorems 9, 22, and 23 are not provided.
- The paper would benefit from a concise explanation of basic biological terminology such as reads, taxa, and reference databases so that readers without a genomics background can better follow the motivation.
- The writing quality and style suggest that parts of the manuscript might have been generated by a language model. If this is the case, the authors should thoroughly revise the manuscript for clarity. The motivation and the presentation of the problem should be improved as well.

---

### Official Review · Reviewer_PeJr · 2025-10-31

**Soundness:** 1
**Presentation:** 1
**Contribution:** 1
**Rating:** 0
**Confidence:** 4

**Summary:**

This paper supposedly describes a novel classification framework hat seeks to replace alignment with hash-based token mapping. This is supposed to be supported by theoretical results for why this approach reduces computational complexity without loss of accuracy.

This paper is seemingly the creation of an LLM. Potential indicators of this are:

1. Section 5.4.1 is empty.
2. The discussion of results in 5.4.2 makes no reference to the included tables.
3. At multiple places, improvements are described using "percentage points" instead of simply using "%".
4. QA-Token is reference at multiple points throughout this paper but this reference is unavailable; the corresponding citation refers to a completely different manuscript.

**Strengths:**

N/A

**Weaknesses:**

N/A

**Questions:**

N/A

**Details Of Ethics Concerns:**

This paper seems to be generated by an LLM or edited heavily with an LLM.

---

### Official Review · Reviewer_kPUY · 2025-11-01

**Soundness:** 1
**Presentation:** 1
**Contribution:** 1
**Rating:** 2
**Confidence:** 4

**Summary:**

The paper proposes HighClass, a metagenomic read classifier that replaces seed-and-extend alignment with hash-based mapping of variable-length, quality-aware tokens and a sparsified inverted index. Algorithmically, it (i) uses a 32k-token QA-BPE-seq vocabulary to extract variable-length tokens, (ii) scores taxa with a quality-weighted log-likelihood, and (iii) prunes the reference with gradient-based sparsification (32% regions retained). Theoretically, it presents uniform convergence bounds $O(\sqrt{V|Y|/n})$, α-mixing concentration for dependent tokens with an explicit variance-inflation factor, and consistency of the maximum-likelihood classifier under identifiability. Empirically, on CAMI II, it reports 85.1% F1 (within 1.5% of state-of-the-art method) with 4.2× speedup and 68% memory reduction (6.8 GB index), plus ablations attributing +6.8 pp to variable-length tokens and +1.9 pp to quality weighting.

**Strengths:**

1. Significance. Demonstrating 4.2× speed with near-SOTA accuracy and 68% memory reduction moves the Pareto frontier and has practical impact for real-time clinical diagnostics and large-scale monitoring scenarios.

**Weaknesses:**

1. The paper mentioned that they enabled the applications of HighClass in edge deployment. However, the 6.8GB memory requirement for deployment mentioned in the paper is not ideal for edge devices.
2. The paper has more than 9 pages for the main text, which is against the policy.
3. Section 5.4.1 is completely missing.
4. The paper cited the paper: From noise to signal: Enabling foundation-model pretraining on noisy, real-world corpora via quality-aware tokenization with full author list and the source as ArXiv. However, I cannot find it on ArXiv but also an anonymous submission to ICLR 2026, which is very concerning.

**Questions:**

1. What is the empirical curve of accuracy/runtime/memory vs. vocabulary size (e.g., 8k/16k/32k/64k)?
2. Were sparsification masks trained on the same references used for evaluation?
3. Does line 307 have a typo?
4. Why is 5.4.1 missing?

---

### Official Review · Reviewer_Xkac · 2025-11-01

**Soundness:** 1
**Presentation:** 1
**Contribution:** 1
**Rating:** 0
**Confidence:** 4

**Summary:**

The paper introduces HighClass, a metagenomic classifier combining variable-length quality-aware tokens, hash-based mapping, and sparsified indexing. It claims to provide both a new theoretical framework for token dependencies and strong empirical efficiency gains.

**Strengths:**

N/A

**Weaknesses:**

1) The “theory” mostly restates standard learning theory with ad hoc constants. The claimed “first comprehensive theory” is not substantiated.

2) Made-up references: arXiv:2406.08251 actually corresponds to a completely different paper (Light-induced fictitious magnetic fields for quantum storage in cold atomic ensembles at https://arxiv.org/pdf/2406.08251), different from the cited one.

3) Some sections are even empty (5.4.1, F1, F2, F3, M7).

**Questions:**

The citation:

*Arvid E. Gollwitzer, Paridhi Latawa, David de Gruijl, Deepak A. Subramanian, and Giovanni Traverso. From noise to signal: Enabling foundation-model pretraining on noisy, real-world corpora via quality-aware tokenization. arXiv preprint arXiv:2406.08251, 2025.*

appears fabricated, as the DOI refers to an unrelated paper: Light-induced fictitious magnetic fields for quantum storage in cold atomic ensembles at https://arxiv.org/pdf/2406.08251. **Could the authors explain how this reference was produced and justify its inclusion?**

**Details Of Ethics Concerns:**

Initially, I found the writing style unusual (some sections even empty and vague theorems), which encouraged me to check the references (there are 19). I noticed the following cited paper:

Arvid E. Gollwitzer, Paridhi Latawa, David de Gruijl, Deepak A. Subramanian, and Giovanni Traverso. From noise to signal: Enabling foundation-model pretraining on noisy, real-world corpora via quality-aware tokenization. arXiv preprint arXiv:2406.08251, 2025.

However, arXiv:2406.08251 actually corresponds to a completely different paper (Light-induced fictitious magnetic fields for quantum storage in cold atomic ensembles at https://arxiv.org/pdf/2406.08251). I also noticed that the cited paper, which is submitted to ICLR this year (https://openreview.net/pdf?id=UDvXiEngkX), exhibits the same writing style and structure, including ambiguous references (e.g., Ming Yu et al. Direct advantage policy optimization. arXiv preprint, 2025, and Xiaowei Yue et al. Value-augmented policy optimization. arXiv preprint, 2025).

---

### Note · Authors · 2026-02-17

I have read and agree with the venue's withdrawal policy on behalf of myself and my co-authors.

---

### Meta-Review · Area_Chair_dMgT · 2026-01-03

**Summary:**

The reviewers and previous auditors questioned whether this article was generated by AI, several issues including but not limited to empty chapters and incorrect or non-existent citations. Therefore, I recommend reject.

**Reviewer Concerns:**

The reviewers and previous auditors questioned whether this article was generated by AI, several issues including but not limited to empty chapters and incorrect or non-existent citations. Therefore, I recommend reject.

**Reviewer Scores:**

All reviewers propose clear reject.

---

### Decision · Program_Chairs · 2026-01-26

Reject